# Perspectives on Mitochondria–ER and Mitochondria–Lipid Droplet Contact in Hepatocytes and Hepatic Lipid Metabolism

**DOI:** 10.3390/cells10092273

**Published:** 2021-09-01

**Authors:** Xiaowen Ma, Hui Qian, Allen Chen, Hong-Min Ni, Wen-Xing Ding

**Affiliations:** Department of Pharmacology, Toxicology and Therapeutics, University of Kansas Medical Center, Kansas City, KS 66160, USA; xma3@kumc.edu (X.M.); hqian@kumc.edu (H.Q.); achen6@kumc.edu (A.C.); hni@kumc.edu (H.-M.N.)

**Keywords:** alcohol, autophagy, lipophagy, lipotoxicity, NAFLD, starvation, steatosis

## Abstract

Emerging evidence suggests that mitochondrion–endoplasmic reticulum (ER) and mitochondrion–lipid droplet (LD) contact sites are critical in regulating lipid metabolism in cells. It is well established that intracellular organelles communicate with each other continuously through membrane contact sites to maintain organelle function and cellular homeostasis. The accumulation of LDs in hepatocytes is an early indicator of non-alcoholic fatty liver disease (NAFLD) and alcohol-related liver disease (ALD), which may indicate a breakdown in proper inter-organelle communication. In this review, we discuss previous findings in mitochondrion–ER and mitochondrion–LD contact, focusing on their roles in lipid metabolism in hepatocytes. We also present evidence of a unique mitochondrion–LD contact structure in hepatocytes under various physiological and pathological conditions and propose a working hypothesis to speculate about the role of these structures in regulating the functions of mitochondria and LDs and their implications in NAFLD and ALD.

## 1. Introduction

Non-alcoholic fatty liver disease (NAFLD) and alcohol-related liver disease (ALD) are two major chronic liver diseases that cause serious health problems and are becoming huge economic burdens worldwide. Both NAFLD and ALD share several common pathological features, ranging from simple steatosis to more severe steatohepatitis, which is associated with increased cell death, inflammation and fibrosis, and can further progress to cirrhosis and even hepatocellular carcinoma (HCC) [1,2,3,4]. Hepatic steatosis, the early feature of both NAFLD and ALD, is characterized by the accumulation of lipid droplets (LDs) in hepatocytes. Several mechanisms may lead to the development of hepatic steatosis, including increased free fatty acid uptake, de novo lipogenesis, triglyceride (TAG) synthesis, decreased fatty acid β-oxidation and very low-density lipoprotein (VLDL) secretion.

Eukaryotic cells have many compartmentalized membrane-bound and membrane-less organelles that perform various essential biochemical reactions. Increasing evidence now suggests that these organelles are not isolated but interconnected in order to facilitate the exchange of small molecules and metabolites through membrane contact sites (MCSs), which are crucial in maintaining organelle function and homeostasis, including lipid homeostasis [5,6,7]. MCSs are referred to as areas of close apposition approximately 10~30 nm between the membranes of two organelles. MCSs have been characterized into the following two types: homotypic (between two of the same organelles) and heterotypic (between two different organelles or two different membrane types) [8]. Heterotypic MCSs involving the endoplasmic reticulum (ER) have been well studied, including the ER–plasma membrane, ER–mitochondria, ER–Golgi, ER–peroxisome, ER–endosome/lysosome and ER–LD. Other MCSs include mitochondria-lysosome, mitochondrial-LD, and LD-lysosome. Increasing evidence suggests that inter-organelle MCSs and the resulting cross-talk play critical roles in regulating lipid metabolism and the progression of NAFLD [6,9,10]. There are many elegant reviews that have summarized MCSs, though we focus only on the possible roles of mitochondria–ER and mitochondria–LD contact sites in hepatocytes in this review [6,7,8]. We also present some ultrastructure electron microscopy (EM) images of hepatocytes from our own studies showing a unique atypical mitochondria–LD contact structure under physiological and pathological conditions and discuss its potential implications in lipid metabolism and NAFLD/ALD. 

## 2. Mitochondria–ER Contact/Mitochondrial-Associated Membrane (MAM)

Mitochondria–ER contact has been reported since the 1950s [11], but it was not until the early 1990s that Jean Vance biochemically isolated a small fraction from rat liver cells containing not only mitochondria but also associated parts of the ER membrane, thereby discovering the existence of phosphatidylserine synthesis enzymes in these isolated structures. She coined the newly discovered structure the mitochondrial-associated membrane (MAM) [12]. Subsequent studies have significantly enriched our understanding of the components and possible functions of mitochondria–ER contact/MAM. The components of mitochondria–ER contact include several mitochondrial outer membrane proteins, such as mitofusin 2 (MFN2), voltage-dependent anion channel (VDAC1) and protein tyrosine phosphatase-interacting protein-51 (PTPIP51). MFN2 is a dynamin-like GTPase that is known to regulate mitochondrial fusion by forming a homotypic dimer and a heterotypic dimer with its paralog MFN1. MFN2 is also enriched at mitochondria–ER contact sites, likely as a tether between the mitochondria and the ER. Several ER integral membrane proteins, such as the B-cell receptor-associated protein 31 (Bap31) and vesicle-associated membrane protein (VAMP)-associated protein B (VAPB), are also critical for mitochondria–ER contact [7,13]. Bap31 is an ER chaperone protein that binds with the mitochondrial outer membrane protein Fis1 in the MAM, which regulates apoptosis [14]. VAPB directly binds with PTPIP51, a mitochondrial outer membrane protein, which forms a complex in the MAM and plays a role in regulating Ca^2+^ homeostasis and autophagy [15].

Calcium (Ca^2+^) homeostasis is critical for hepatocyte functions. High concentrations of Ca^2+^ are stored in the ER (ranges from 100–800 µM), but the ER of Ca^2+^ can be released in response to the activation of specific receptors on the ER surface, including inositol 1,4,5-trisphosphate receptors (IP3R) and ryanodine receptors (RyR). The released Ca^2+^ is taken up by mitochondria through voltage-dependent anion channels (VDACs) in the outer mitochondrial membrane at MAMs. Glucose-regulated protein 75 (GRP75) is a heat shock protein and acts as bridging protein for the physically interactions of IP_3_R with VDAC. The mitochondrial calcium uniporter (MCU) regulates Ca^2+^ uptake across the inner mitochondrial membrane into the matrix,   which is driven by the negative charge of mitochondrial membrane potential [16,17]. While the physiological levels of mitochondria Ca^2+^ is important in regulating mitochondria metabolism, persistent increased mitochondrial Ca^2+^ levels can lead to the opening of the mitochondrial permeability transition pore and apoptosis. Dysregulation of hepatic Ca^2+^ homeostasis has been linked to NAFLD/NASH, as the increased expression of IP3R is associated with the degree of steatosis in NASH patients [18,19].

In primary cultured mouse hepatocytes, mitochondria–ER contact sites are readily detected under electron microscopy (EM) analysis (Figure 1A,B, white arrows). Almost all of the mitochondria in a hepatocyte are in close contact with the ER membrane, with some even being almost entirely wrapped by ER (Figure 1C, white arrow). In addition, ER membranes are also found in close contact with LDs (Figure 1A,B, black arrows). These data suggest that there are abundant mitochondria–ER and ER–LD contact sites in hepatocytes, reflecting their crucial roles in maintaining hepatocyte function, which will be discussed below. 

Mitochondria–ER contact sites have a variety of biological functions. Mitochondria are dynamic organelles that constantly undergo fission and fusion [20]. One role of ER–mitochondria contact is to mark sites on the mitochondria for fission by constricting and recruiting mitochondrial fission promotor protein dynamin-related protein 1 (Drp1), which is also a dynamin-like GTPase [21]. Mitochondria–ER contact sites also coordinate the calcium transfer between ER and mitochondria via the physical interaction between VDAC1 and IP3R, the ER calcium release channel [22,23]. A recent study reported the contact of rough ER with mitochondria in hepatocytes, which is termed as wrappER–mitochondria contact [24]. The wrappER–mitochondrial contact site may serve as a site for VLDL biogenesis in the mouse liver, and ablation of the wrappER–mitochondrial contact decreases the VLDL secretion, resulting in the systemic changes in lipids in mice [24].

Perhaps one of the most characterized functions of the MAM is its role in lipid metabolism. Phospholipids (PLs), major components of biological membranes, are composed of two hydrophobic fatty acyl chain “tails” and one hydrophilic “head” group. PLs have many important functions, being particularly important in segregating cellular contents and forming organelles. They also serve as substrates for the generation of a variety of signaling molecules, such as lysophosphatidylcholine (LPC), lysophoshatidic acid (LPA) and diacylglycerol [25,26,27]. The major structural PLs in the mammalian membrane are phosphatidylcholine (PC), phosphatidylethanolamine (PE), phosphatidylserine (PS), phosphatidylinositol (PI) and phosphatidic acid (PA). Among them, PC and PE are the two most abundant PLs in mammals, both as part of the cell membrane and within lipoproteins, accounting for 70–80% of total PLs [28,29,30]. The Kennedy pathway is one of the major pathways for the biosynthesis of PC and PE [30]. Choline and ethanolamine enter cells and perform a series of reactions to synthesize PC and PE. Two enzymes located in the ER, choline phosphotransferase (CPT) and choline/ethanolamine phosphotransferase (CEPT), are used in the last step in PC and PE synthesis, respectively. In addition to the Kennedy pathway, the liver also utilizes another pathway for PC synthesis. In this three-step pathway, PE is methylated to PC by phosphatidylethanolamine N-methyltransferase (PEMT) at the MAM. Another pathway for PE synthesis occurs in the mitochondrial inner membrane, mediated by phosphatidylserine decarboxylase 1 (PSD1). PS synthesized at the MAM by phosphatidylserine synthase (PSS) is transported to the mitochondria, where PSD1 is located. PE can then be quickly transported to the ER and other membranes. PLs have been shown to be involved in lipid metabolism, including LD formation, lipoprotein production and VLDL secretion [25,31]. A lack of PC and PE biosynthesis, either by a controlled diet (methionine- and choline-deficient diet) or a genetic knockout (PEMT, Pcyt1α, Pcyt2 etc.), causes impaired VLDL assembly and secretion, hepatic steatosis and hypolipidemia under both fasting and normal conditions [32,33,34,35,36,37]. A change in PC and/or PE content in various tissues is implicated in NAFLD and metabolic disorders, such as atherosclerosis, insulin resistance and obesity [30]. A PL remodeling process known as the Lands cycle modifies the composition and asymmetrical distribution of fatty acyl chains in PLs after their de novo synthesis [38]. The fatty acyl moieties of membrane PLs determine the biophysical properties of the cell membrane, including the fluidity, curvature and subdomain architecture, which in turn affects vesicle trafficking, signal transduction and molecular transport [25,39]. PLs contain polyunsaturated fatty acyl-chains; the incorporation of PLs into membranes might be able to facilitate the transfer of lipids from the ER bilayer to ApoB in the ER lumen. Therefore, not only the PC and PE amounts, but also their fatty acyl chain composition, especially with regards to the linoleoyl and arachidonyl chains, have been shown to regulate VLDL secretion [40,41]. The deletion of lysophosphatidylcholine acyltransferase 3 (Lpcat3), a target gene of the liver X receptor (LXR), reduces the amounts of PC, PE and acyl chains, which in turn cause a low ER membrane curvature and VLDL secretion [40]. In addition to PL synthesis enzymes, MAMs also contains other critical lipid metabolism enzymes. Fatty acid CoA ligase 4 (FACL4), an enzyme that regulates the ligation of fatty acids to coenzyme A (CoA), has been used as a common MAM marker [42]. Moreover, acyl-coA:cholesterol acyltransferase-1 (ACAT1/SOAT1) and acyl-CoA:diacylglycerol acyltransferase 2 (DGAT2), enzymes that regulate the esterification of cholesterol and triglyceride synthesis, are also enriched in MAMs [43,44]. The pathways and mitochondria–ER contact sites involved in regulating PL synthesis are summarized in Figure 2. 

The Kennedy pathway enzymes are expressed ubiquitously in eukaryotes in order to regulate the synthesis of PLs, which are essential to form cellular membranes to protect the cells and make barriers for organelles within the cells. In hepatocytes, the dysregulation of PLs synthesis, in particular the abundance of PC and PE, as well as the molar ratio of PC/PE, can affect the size and dynamics of LDs and assembly and secretion of VLDL, as well as mitochondrial bioenergetics [30]. For instance, liver-specific Pcyt1α (encoding the protein CT) knockout mice and whole-body Pemt knockout mice fed with a high fat diet have decreased PC contents in hepatocytes and impaired hepatic VLDL secretion [34,45,46]. In addition, the hepatocytes’ lack of PEMT has decreased the PC/PE molar ratio with an increased mitochondrial PE content, which leads to a reduced hepatic glucose production from pyruvate and increased accumulation of elongated mitochondria and mitochondrial respiration, as well as ATP production [47]. As a result, Pemt knockout mice are resistant to high-fat-diet-induced obesity and insulin resistance, although they still develop NASH [45]. 

To date, there is still a very limited understanding on the local concentration and actual lipid composition of the MAMs. It has been shown that the depletion of cholesterol in MAMs with methyl-β-cyclodextrin (MβC) increases the association of MAMs with mitochondria. It is likely that there are different protein and lipid compositions in MAMs in different cell types, or in response to different physiological or pathological stresses. It is also perceivable that the changes in the specific lipid contents and combinations within the MAMs may lead to the recruitment of different sets of proteins, and thus the functions of MAMs. Future works are needed to specifically manipulate the lipid contents and compositions in the MAMs in order to further dissect the role of PLs in regulating MAMs and its subsequent impacts on ER and mitochondrial functions.

Mitochondria–ER contact sites have been shown to be the origin of the autophagsome membrane in mammalian cells [48]. However, it has long been known that the growth of autophagosomes requires adding lipids for membrane elongation, which requires a protein-mediated lipid transfer process. ATG2 localizes to contact sites between the nascent autophagosome and the ER and mediates lipid transfer from the ER to the autophagosome membrane [49]. The transfer of lipids occurs only between the cytosolic leaflets of the opposed bilayers; thus, lipids need to be equilibrated between the inner and outer leaflets of both the membranes of the lipid donor and the lipid acceptor. The equilibration of lipids between the membranes is mediated by a scramblase, a protein that is responsible for the translocation of phospholipids between the two monolayers of a lipid bilayer [50]. Vacuole membrane protein 1 (VMP1) and transmembrane protein 41B (TMEM41B) are both ER transmembrane proteins, which play critical roles in LD biogenesis, VLDL assembly and secretion and regulating the formation and closure of autophagosomes [51,52,53]. Recent evidence indicates that both VMP1 and TMEM41B, as well as ATG9, have lipid scramblase activity that regulate the shuttle of phospholipids across the membrane–lipid bilayer [54,55,56], which may promote autophagosome membrane expansion and eventual formation. In addition to regulating autophagosome formation, VMP1 also regulates organelle contacts, including ER–mitochondrial and ER–autophagosome contact [57,58]; though whether VMP1-mediated ER–mitochondrial contact contributes to VLDL assembly and secretion remains unknown. The loss of hepatic TMEM41B in mice results in severe hepatic steatosis due to impaired VLDL secretion and increased de novo lipogenesis, revealing the crucial role of the protein in these cellular pathways [54]. 

The vacuolar protein sorting-associated protein 13 (VPS13) family contains the following four members in humans: VPS13A, B, C and D. VPS13A and VPS13C are lipid transporters at the ER–mitochondria, ER–LD and ER–late-endosome contact sites [59], whereas VPS13B regulates the Golgi integrity [60]. Human VPS13A and VPS13C are both involved in ER–organelle contact, with VPS13A tethering ER to mitochondria and VPS13C tethering ER to late endosomes/lysosomes. Both VPS13A and VPS13C mediate ER–LD tethering [59]. Unlike VPS13A-C, VPS13D contains an ubiquitin binding (UBA) domain, which is required for regulating mitochondrial size and clearance. Cells that express mutant VPS13D lacking the UBA domain have a defective mitochondrial clearance and thus accumulate giant mitochondria [61]. VPS13D interacts with the mitochondrial fusion protein Marf (mitofusins in mammals) in *Drosophila* and functions in a pathway downstream of VMP1 and upstream of Marf/MFN2 in regulating mitochondrial–ER contact and autophagy [58]. The outer mitochondrial membrane GTPase protein Miro, and its peroxisome-enriched splice variant, bind to VPS13D, providing a lipid conduit between the ER and mitochondria [62]. In addition to regulating mitochondrial morphology, VPS13D-deficient cells have a partial or complete loss of peroxisomes, suggesting VPS13D is required for peroxisome biogenesis [63]. Interestingly, VPS13D interacts with the endosomal sorting complex required for transport (ESCRT) protein tumor susceptibility 101 (TSG101). VPS13D and TSG101 coordinately regulate the trafficking of the free fatty acid (FFA) between mitochondria and LDs at contact sites in cultured cells. The depletion of VPS13D or TSG101 inhibits FFA trafficking from LDs to the mitochondria based on a BODIPY-conjugated FFA assay [64]. These recent findings demonstrate the critical roles of VPS13 family proteins in regulating organelle contact, dynamics and biogenesis, as well as autophagy and lipid metabolism, which are vital for cellular function and homeostasis. Indeed, mutations in VPS13 family proteins have been associated with various recessive movement disorders. Notably, loss-of-function mutations in VPS13D cause a type of autosomal recessive ataxia with an abnormal mitochondrial morphology [65,66,67]. However, the role of VPS13 family proteins in regulating mitochondria and LDs in the context of liver diseases remains to be studied. 

## 3. Lipid Droplets (LDs) and Autophagy

LDs are intracellular dynamic organelles with a wide range of sizes, from tens of nm to several µm in diameter, that primarily serve to store triacylglycerol (TAG) and sterol esters as a bioenergy source [68]. Despite extensive studies, precisely how LDs are formed remains obscure. Nonetheless, current evidence points towards LDs either being formed from by de novo synthesis or via the fission of existing LDs. LDs are thought to originate from the ER through at least four discrete stages: the synthesis and accumulation of neutral lipids within the ER bilayer, formation of a lens structure, budding from the ER into the cytosol and growth of LDs [68,69,70]. 

Under starvation conditions, cells shift their metabolism from glycolysis to fatty acid oxidation as an adaptive response. This is mainly achieved by breaking down LDs and then transporting FFA to mitochondria for β-oxidation. Cells also activate autophagy as another adaptive response to starvation, causing them to degrade cellular membrane organelles and proteins in order to generate biomolecules (amino acids and FFA) for survival. However, FFAs are toxic to cells, generating reactive oxygen species (ROS) that can damage mitochondria, triggering cell death via lipotoxicity [71,72,73]. The excess FFAs generated by autophagy can either be channeled to form LDs or to mitochondria to undergo β-oxidation in order to avoid inducing lipotoxicity [74,75]. Notably, in addition to using intracellular lipid stores, cells also respond to starvation by increasing the uptake of extracellular FFAs. In humans and mammals, most extracellular FFAs are secreted from adipose tissue following increased lipolysis due to reduced levels of circulating glucose and insulin under fasting conditions. This process also leads to hepatic steatosis due to the increased uptake of FFAs by hepatocytes [76,77,78,79]. Moreover, fasting also causes elevated levels of catecholamines in the blood and cAMP in adipose cells, which activates protein kinase A (PKA), resulting in the increased phosphorylation of several LD-associated proteins, including perilipin 1, hormone-sensitive lipase (HSL) and comparative gene identification-58 (CGI-58). CGI-58 acts as a coactivator that increases the catalytic activity of patatin-like phospholipase domain containing 2 (PNPLA2)/adipose triglyceride lipase (ATGL), promoting the sequential hydrolysis of TAG into DAG, which then further converts to monoacylglycerol (MAG) by HSL, and eventually into non-esterified FFAs and glycerol by monoacylglycerol lipase (MGI) [80]. Increased circulating/extracellular FFAs derived from lipolysis within adipose tissue then enter cells, including hepatocytes, via plasma membrane fatty acid transporters, where they are either oxidized to generate acetyl-CoA and ketone bodies to produce energy, or are stored in LDs, which occurs in the liver [81,82]. As discussed above, excessive FFAs are toxic; therefore, cells must avoid cytotoxicity by employing safe, efficient routes to redirect FFAs to either form LDs or to mitochondria for beta-oxidation. How FFAs are channeled from LD lipolysis to mitochondria remains unknown, but it may involve the formation of mitochondria–LD contact sites. 

## 4. Mitochondria–LD Contact

Mitochondria–LD contact has been reported in a variety of cell types and tissues, although the molecular machinery that regulates the contact is just beginning to be uncovered. Several key players that are known to regulate mitochondria–LD contact include the SANRE protein SNAP23, mitochondrial fusion protein MFN2 and a mitochondria outer membrane protein mitoguardin 2 (MIGA2), as well as the LD protein perilipin 1 (PLN1) and perilipin 5 (PLN5) [83]. SNAP23 may interact with the long chain acyl-CoA synthetase 1 (ACSL1), which is located on the mitochondrial surface, to promote the mitochondria–LD contact [84]. The knockdown of SNAP23 inhibits mitochondria–LD contact and decreases β-oxidation [85]. In brown adipose tissue, PLN1 directly interacts with MFN2 and the ablation of MFN2 decreases mitochondria–LD contact [86]. PLN5 is also detected on mitochondria–LD contact sites and promotes the contact formation, although its interacting partner on mitochondria has not yet been identified [87]. MIGA2 also directly tethers mitochondria to LD via its amphipathic LD-targeting motif and both promotes de novo lipogenesis in mitochondria and facilitates lipid storage in LDs [88]. The currently known machinery that regulates mitochondria–LD contact is illustrated in Figure 3. 

After TAG break down, either through TAG hydrolysis by cytosolic lipases, such as PNPLA2/ATGL, or through lysosomal acidic lipases via lipophagy [74,89,90], the released FFAs are then utilized to produce ATP by β-oxidation, the citric acid cycle and oxidative phosphorylation, processes in which mitochondria play key roles. The accumulation of LDs in hepatocytes is a well-known early feature of NALFD and ALD. In primary cultured mouse hepatocytes that were treated with ethanol, the increased accumulation of LDs was readily detected, with some of the LDs in close proximity to the mitochondria (Figure 4). Interestingly, we observed an electron-dense membrane-enriched structure, either in a dot or tubular shape, within LDs in close proximity to the mitochondria (Figure 4, arrows). The atypical mito–LD contact was also observed in regular cultured primary hepatocytes, although the frequency seemed to be much lower than ethanol-treated or starved hepatocytes (Figure 4A, arrow). It is likely that the atypical “mito–LD” contacts would increase in conditions with a high energy demand. More careful and rigorous quantitative EM work is needed in the future to further confirm whether the atypical mito–LD contact would be increased in response to a specific condition. Similar structures were also observed in hepatocytes that underwent starvation or that were treated with the unsaturated fatty acid oleic acid (OA) (Figure 5, white arrows). In addition to mitochondria, peroxisomes are another major site for FFA β-oxidation. Fittingly, we also observed the close contact of peroxisomes with LDs in OA-treated hepatocytes (Figure 5, red arrows). Notably, mitochondria–LD and peroxisome–LD contacts are also observed in Atg5-deficient hepatocytes (Figure 5, red arrows), suggesting that autophagy may be unnecessary in the formation of these interactions. Similar electron-dense membrane structures were also observed in the livers of mice that were fasted overnight (Figure 6, arrows). It is currently unknown how these previously undescribed electron-dense membrane structures are formed in mitochondria–LD contact sites. It is possible that the membrane structures could be derived from the protrusion of the mitochondrial membrane into the LDs or de nova synthesis at the contact sites. Future studies are needed to confirm these mitochondrial–LD contacts and determine their origin. Moreover, the pathophysiological significance is also unclear, although it may have several possible implications that we would like to hypothesize on below.

First, as discussed above, FFAs in the cytosol are toxic to the cells; thus, we conceive that the formation of a direct channel to transport FFAs from LDs (lipid storage) to mitochondria (FFAs oxidation/utilization) would be an efficient way to avoid high cytosolic concentrations of FFAs. Second, in addition to serving as a conduit channel, it is likely that these electron-dense membrane structures may contain several key proteins that function to tether the two organelles together and stabilize the mitochondria–LD contact sites. Perilipin 5, a LD structural protein, and synaptosome-associated protein 23 (SNAP23), a SNARE protein, are two such proteins that have been reported to regulate mitochondria–LD contact [85,87]. Future studies that would employ immune gold EM are needed to determine whether perilipin 5 and SNAP23 are located on these electron-dense structures. In brown adipocytes, perilipin 1 directly interacts with MFN2, but not MFN1, to regulate mitochondria–LD contact in response to adrenergic stimulation [86]. However, whether MFN2 and perilipin 1 also regulate mitochondria–LD contact in hepatocytes remains to be determined. Third, it is also likely that these electron-dense structures may serve as a recognition/targeting site to initiate mitochondria–LD contact. Fourth, mitochondria are one of the major sites for the synthesis of phospholipids, such as PE, which can be further converted to PC, the main LD phospholipid [91,92]. In contrast to the FFA transfer from the LD to the mitochondria for energy production, mitochondria–LD contact may also transfer lipids from the mitochondria to LDs. Fifth, both mitochondria and LDs are highly dynamic. It is likely that the mitochondria–LD contact site may be enriched with mitochondria fusion (MFN1 and MFN2) and fission proteins (such as Drp1), as well as mitochondria motility proteins (such as Miro) that act as markers for mitochondrial fission or fusion and movement [20,93]. Conversely, mitochondrial–LD contact sites may support LD expansion by increasing ATP-synthase-dependent TAG synthesis or by promoting LD trafficking [94]. Therefore, it is likely that electron-dense mitochondria–LD contact sites may be important for both lipolysis and lipogenesis in response to different metabolic demands. It should also be noted that the electron-dense structures that we described here are morphologically distinct from the structures found in previously reported LD-anchored mitochondria (LDAM) or peridroplet mitochondria (PDM) identified in oxidative tissues, such as brown adipose tissue and muscle, as they did not contain electron-dense membranes [94,95,96]. 

## 5. Conclusions and Future Directions

In summary, organelle communication through membrane contact sites has emerged as a hot biomedical research topic, as improved biochemical and imaging techniques allow for the identification of molecular components and the visualization and quantification of contact sites. In the event of mitochondria contact with other organelles, such as the ER and LDs, many questions remain unanswered. While many components in the mitochondria–ER and mitochondria–LD contact sites have been identified, most of these studies were carried out in isolated organelles or in vitro cultured cells; the physiological relevance of these proteins in vivo still remains largely obscure. Given the importance of mitochondria–ER and mitochondria–LD contact in regulating lipid metabolism, further studies using relevant animal models are needed to investigate how the loss or impairment of these contacts would affect hepatocyte lipid homeostasis and NAFLD or ALD. 

## Figures and Tables

**Figure 1 cells-10-02273-f001:**
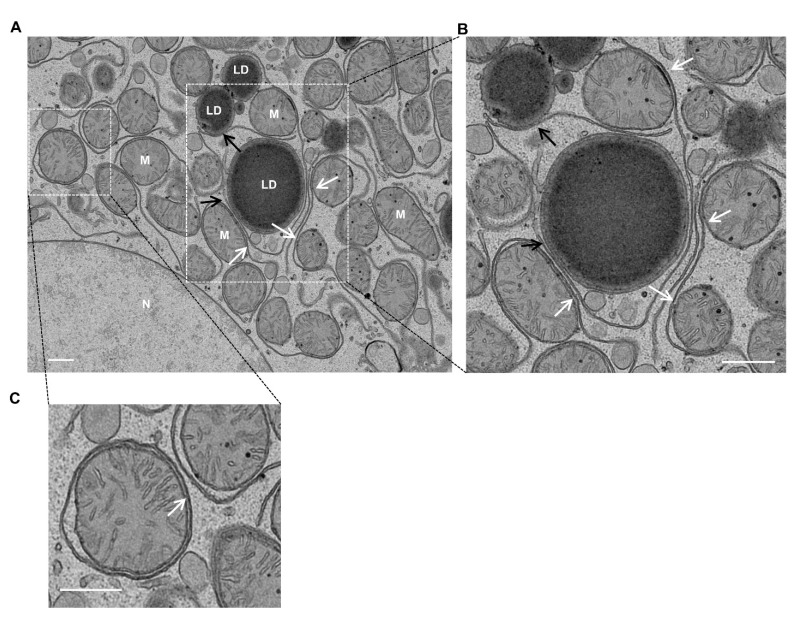
Ultrastructure illustrations of membrane contacts in hepatocytes. Primary mouse hepatocytes were cultured for 24 h, then analyzed by EM. (**A**) Representative EM image of a primary cultured hepatocyte. (**B**,**C**) Enlarged images from the boxed areas in A. White arrows denote mitochondria–ER contact and black arrows denote ER–LD contact. LD: lipid droplet; M: mitochondria; N: nucleus. All of the EM imaging data in this manuscript are from the authors’ lab and have not been previously published. Bar: 500 nm.

**Figure 2 cells-10-02273-f002:**
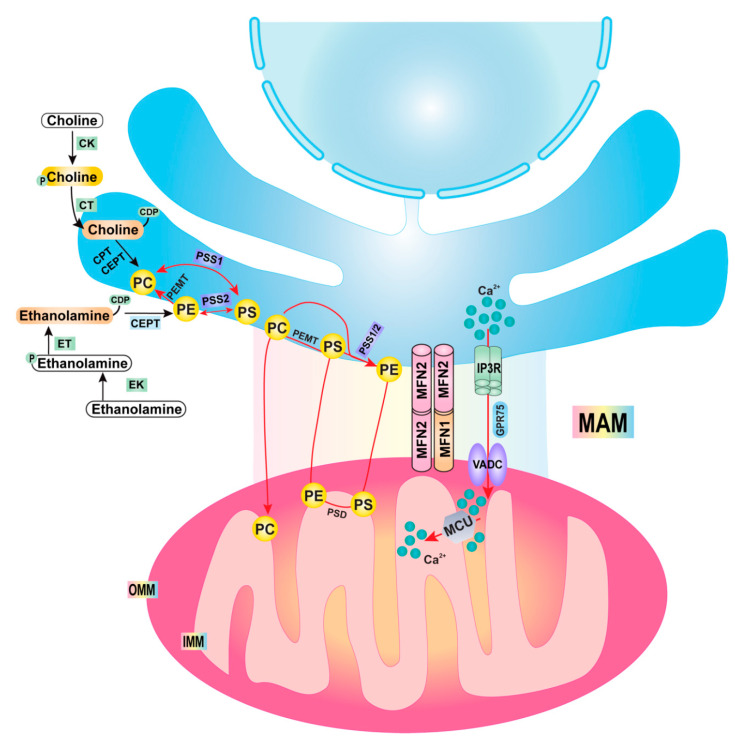
Phospholipid synthesis pathways and membrane contacts. There are two branches of the Kennedy pathway: the CDP-choline and the CDP-ethanolamine pathways. Imported extracellular choline is first phosphorylated by CK (a cytosolic enzyme) to phosphocholine, then converted to CDP-choline by CT (a nuclear and cytosol enzyme), before finally being catalyzed to PC by CPT (an integral ER membrane protein) by transferring a diacylglycerol to CDP-choline. For PE synthesis, ethanolamine is first phosphorylated to phosphoethanolamine by EK (a cytosolic protein), then converted to CDP-ethanolamine by ET (another cytosolic enzyme), before finally being catalyzed to PE by EPT (also an integral ER membrane protein) by transferring a diacylglycerol to CDP-ethanolamine. Alternatively, PE can also be converted to PC by three successive methylation reactions catalyzed by PEMT (an ER/MAM enzyme). In addition, PS is transported to mitochondria and decarboxylated to PE by PSD (an enzyme on the mitochondrial inner membrane). For PS synthesis, PSS1(an enzyme on MAM) catalyzes the exchange of choline for serine in PC, whereas PSS2 (also on MAM) catalyzes the exchange of ethanolamine for serine in PE. Enzymes of the ER membrane are indicated in blue, whereas cytosolic enzymes are in red. Enzymes: CK, choline kinase; CCT, phosphocholine cytidylyltransferase; CPT, cholinephosphotransferase; CTP, phosphocholine cytidylyltransferase (CT); EK, ethanolamine kinase; ECT, phosphoethanolamine cytidylyltransferase; EPT, ethanolaminephosphotransferase; PS decarboxylase (PSD); PS synthase-1 (PSS1); PS synthase-2 (PSS2); MAM: mitochondrial associated membrane; MFN1: mitofusin 1; MFN2: mitofusin 2; IP3R: inositol 1,4,5-trisphosphate receptor; GRP75: glucose-regulated protein 75; MCU: mitochondrial calcium uniporter; VDAC: voltage-dependent anion channel.

**Figure 3 cells-10-02273-f003:**
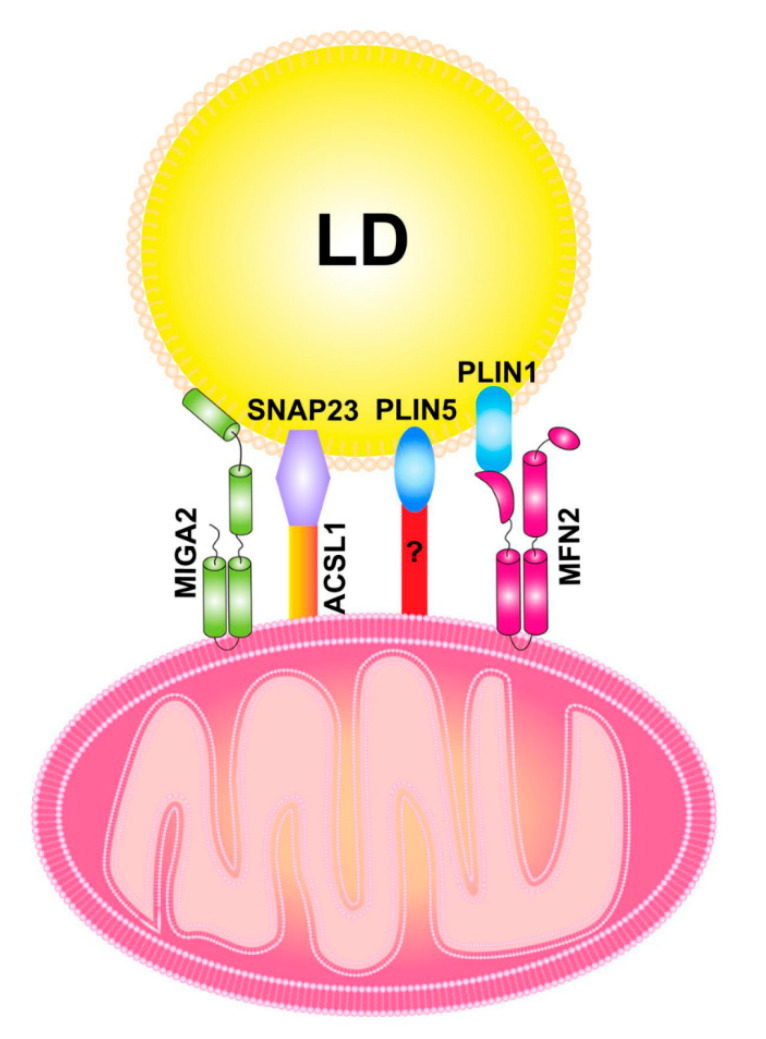
A proposed model of mitochondria–LD contact machinery. PLN1 and PLN5 are LD surface proteins that are implicated in mitochondria–LD contact. PLN1 interacts with MFN2 to promote the tethering of mitochondria with LD. The binding partner for PLN5 is currently unknown. SNAP23 interacts with ACSL1, which is located at outer mitochondrial membrane to promote mitochondria–LD contact. MIGA2, an outer mitochondrial membrane protein, directly tethers mitochondria with LD. ACSL1: acyl-CoA synthetase 1; LD: lipid droplet; MFN2, mitofusin 2; PLN1, perilipin 1; PLN5: perilipin 5; SNAP23: synaptosome-associated protein 23.

**Figure 4 cells-10-02273-f004:**
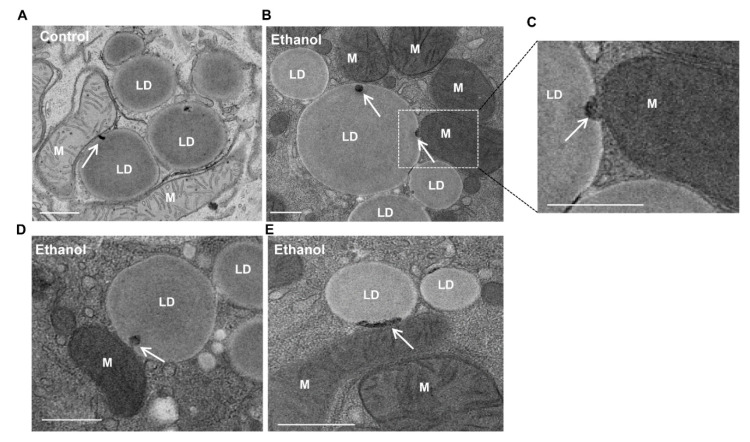
Dot- or tubular-like mitochondria–LD contact sites in primary cultured mouse hepatocytes. Primary mouse hepatocytes were either untreated (**A**) or treated with ethanol (100 mM, **B**–**E**) for 6 h followed by EM analysis. (**A**–**E**) Representative EM images from untreated control hepatocytes (**A**) or ethanol-treated hepatocytes (**B**–**E**) are shown. (**C**) Enlarged image from the boxed areas in B. White arrows denote electron-dense dot- and tubular-like mitochondria–LD contact sites. LD: lipid droplet; M: mitochondria. Bar: 500 nm.

**Figure 5 cells-10-02273-f005:**
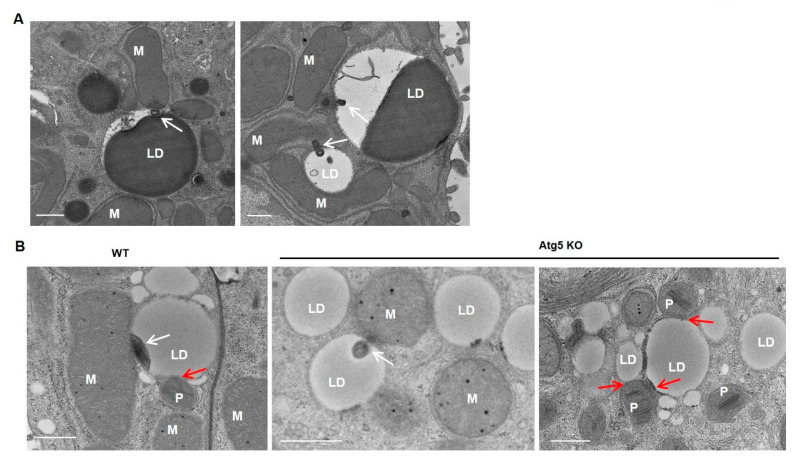
Dot- or tubular-like mitochondria–LD contact sites in starved or oleic-acid-treated primary mouse hepatocytes. Primary mouse hepatocytes were cultured in EBSS for 2 h followed by EM analysis. (**A**) Representative EM images from hepatocytes cultured in EBSS are shown. Hepatocytes isolated from liver-specific Atg5 knockout (KO) mice and the matched wild-type (WT) mice were treated with oleic acid (OA, 500 µM) for 6 h followed by EM analysis. (**B**) Representative EM images from OA-treated WT hepatocytes and Atg5 KO hepatocytes are shown. White arrows denote electron-dense dot- and tubular-like mitochondria–LD contact sites. Red arrows denote peroxisome–LD contact. LD: lipid droplet; M: mitochondria; P: peroxisome. Bar: 500 nm.

**Figure 6 cells-10-02273-f006:**
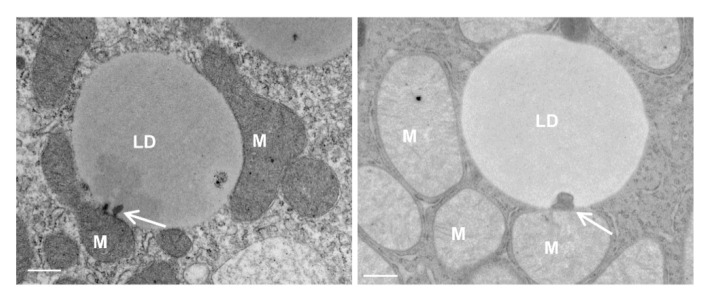
Dot-like mitochondria–LD contact sites in fasted mouse livers. C57Bl/6J mice were fasted overnight and liver tissues were collected followed by EM analysis (experimental details see ref. [67]). Representative EM images from mouse livers are shown. White arrows denote electron-dense dot-like mitochondria–LD contacts. LD: lipid droplet; M: mitochondria; P: peroxisome. Bar: 500 nm.

## Data Availability

Not applicable.

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
