# Peer review of "Perspectives on Mitochondria–ER and Mitochondria–Lipid Droplet Contact in Hepatocytes and Hepatic Lipid Metabolism"

_cells, 2021, doi:10.3390/cells10092273_

Round 1

Reviewer 1 Report

I only have minor concerns

In the second section (2. Mitochondria-ER Contact/Mitochondrial-Associated Membrane (MAM)), authors introduce many proteins and their abbreviation without referring to it at other place in the text and without putting them into perspective in the context of membrane contact (eg: PTPIP51, Bap31, VAPB). It make the text unnecessarily difficult to read and follow. Please simplify

Again, in the second section, the synthesis pathway of phospholipids could be summarized further, and more emphasis should be placed on the effect of those phospholipids change in hepatocytes as well as on MAM dynamism.

In the third section at line 217-221, with the reference presented, I am not convinced that FFA secreted from adipose tissue during starvation leads to hepatic steatosis. This directly link between FFA and hepatic need to be supported by previous published studies

The fourth section is very clear and well organized.

Overall this is a good review linking the hypothesis of LD-mitochondria contact site as a strategy to facilitated FFA beta-oxidation while minimizing FFA cytotoxicity.

Author Response

R#1: I only have minor concerns

In the second section (2. Mitochondria-ER Contact/Mitochondrial-Associated Membrane (MAM)), authors introduce many proteins and their abbreviation without referring to it at other place in the text and without putting them into perspective in the context of membrane contact (eg: PTPIP51, Bap31, VAPB). It make the text unnecessarily difficult to read and follow. Please simplify.

Response to Comment #1: Thanks for the comment. We agree with this reviewer that some more information should be provided for PTPIP51, Bap31, VAPB. As these are critical MAM components, we think it would be better to keep them in the text to allow readers to have a more comprehensive view on MAM. Therefore, we added more information for PTPIP51, Bap31, VAPB in the revised manuscript as below:

“Bap31 is an ER chaperone protein that binds with the mitochondrial outer membrane protein Fis1 in the MAM, which regulates apoptosis.  VAPB directly binds with PTPIP51, a mitochondrial outer membrane protein, which forms a complex in the MAM and plays a role in regulating Ca2+ homeostasis and autophagy”.

Again, in the second section, the synthesis pathway of phospholipids could be summarized further, and more emphasis should be placed on the effect of those phospholipids change in hepatocytes as well as on MAM dynamism.

Response to Comment #2: Thanks for the comment. We agree with the reviewer and has extended our discussions on the PLs synthesis in hepatocyte functions and potential effects on MAMs as below:

“The Kennedy pathway enzymes are expressed ubiquitously in eukaryotes to regulate the synthesis of PLs, which are essential to form cellular membranes to protect the cells and make barriers for organelles within the cells. In hepatocytes, dysregulation of PLs synthesis, in particular the abundance of PC and PE as well as the molar ratio of PC/PE can affect the size and dynamics of LDs, assembly and secretion of VLDL as well as mitochondrial bioenergetics. For instance, liver-specific Pcyt1α (encoding the protein CT) knockout mice and whole body Pemt knockout mice fed with a high fat diet have decreased PC contents in hepatocytes and impaired hepatic VLDL secretion. In addition, hepatocytes lack of PEMT have decreased PC/PE molar ratio with increased mitochondrial PE content, which leads to reduced hepatic glucose production from pyruvate, increased accumulation of elongated mitochondria and mitochondrial respiration as well as ATP production. As a result, Pemt knockout mice are resistant to high fat diet-induced obesity and insulin resistance although they still develop NASH.

To date, there is still very limited understanding on the local concentration and actual lipid composition of the MAMs. It has been shown that depletion of cholesterol in MAMs with methyl-β-cyclodextrin (MβC) increases the association of MAMs with mitochondria.  It is likely that there are different proteins and lipid compositions in MAMs in different cell types or in response to different physiological or pathological stresses. It is also perceivable that the changes of the specific lipid contents and combinations within the MAMs may lead to recruit to different sets of proteins and thus the functions of MAMs. Future works are needed to specifically manipulate the lipid contents and compositions in the MAMs to further dissect the role of PLs in regulating MAMs and its subsequent impacts on ER and mitochondrial functions”.

In the third section at line 217-221, with the reference presented, I am not convinced that FFA secreted from adipose tissue during starvation leads to hepatic steatosis. This directly link between FFA and hepatic need to be supported by previous published studies.

Response to Comment #3: Thanks for the comment. Starvation/fasting induces hepatic steatosis has been well documented in the literature (PMID: 10844002; 17904417; 33513687) and we added more relevant references as suggested. However, we agree with this reviewer this may not be the only mechanism that induce hepatic steatosis during starvation.

The fourth section is very clear and well organized.

Overall this is a good review linking the hypothesis of LD-mitochondria contact site as a strategy to facilitated FFA beta-oxidation while minimizing FFA cytotoxicity.

Response to Comment #4: Thanks so much for the valued input and appreciate the positive comment.

Reviewer 2 Report

The article summarizes current topics in mito-LD and mito-ER contacts concisely. Moreover, they report an atypical type of mitochondria-LD contacts by showing electron micrographs of mouse hepatocytes after ethanol, OA treatment or in starvation.

Following points should be considered prior to publication.

  1. They propose a model of mito-LD contacts in hepatocytes in Fig. 5B. Unfortunately, no evidence was provided to support the model. Even if it is clearly stated that the model is speculative, the model can mislead readers. Therefore, I would suggest following. i) The authors shall delete Fig. 5B; ii) insert a paragraph describing known components of mito-LD contacts, including MIGA2 in the beginning of “4. Mitochondria-LD Contact”; ii) show a figure depicting these contacts but not specific to hepatocytes (visually it will be very similar to Fig. 5B) as Fig. 3; C) and then come to the description of the new type of mito-LD contacts found in hepatocytes.
  2. When I saw the electron micrographs of mito-LD contacts (especially Fig. 3B, D or 5A), I had a strong feeling that a part of mitochondria protrudes into LD and thus the contacts look electron-dense. I would like to hear author’s opinion on this view.
  3. Are these atypical mito-LD contacts seen only after treatment or starvation? The authors should include micrographs from control (non-treated or mock) hepatocytes to make clear this point.
  4. Honestly saying, I could not see any direct physiological link between progression of NAFLD/ALD and mito-LD contacts, although there is a link between NAFLD/ALD and LD formation/beta oxidation, or between LD formation/beta oxidation and mito-LD contacts.
  5. Calcium homeostasis and the role of mito-ER contact have been implicated in the progression of NAFLD. Although it is somewhat depicted in the Fig. 2, I could not find any text mentioning the physiological link between NAFLD and calcium homeostasis. The authors should describe it somewhere in the text.

Author Response

Reviewer #2:

The article summarizes current topics in mito-LD and mito-ER contacts concisely. Moreover, they report an atypical type of mitochondria-LD contacts by showing electron micrographs of mouse hepatocytes after ethanol, OA treatment or in starvation.

Following points should be considered prior to publication.

  1. They propose a model of mito-LD contacts in hepatocytes in Fig. 5B. Unfortunately, no evidence was provided to support the model. Even if it is clearly stated that the model is speculative, the model can mislead readers. Therefore, I would suggest following. i) The authors shall delete Fig. 5B; ii) insert a paragraph describing known components of mito-LD contacts, including MIGA2 in the beginning of “4. Mitochondria-LD Contact”; ii) show a figure depicting these contacts but not specific to hepatocytes (visually it will be very similar to Fig. 5B) as Fig. 3; C) and then come to the description of the new type of mito-LD contacts found in hepatocytes.

Response to Comment #1: Thanks for the comment. We fully agree with this reviewer that the proposed “mito-LD” contact we observed is speculative. This is our intention to share these observations with other colleagues and researchers, and hopefully stimulate more works by others to further confirm the observation and characterize the role of “mito-LD” contact in more physiological and pathological settings. Nonetheless, we have made changes as suggested by this reviewer to discuss the general “Mito-LD” contact first followed by discussing our own findings and perspectives on these findings to make it more clear and avoid potential misleading.

As suggested, we also have divided Figure 5 into two figures with Figure 5B as the new Figure 3 and Figure 5A as the new Figure 6. We also added a new paragraph to discuss the general “Mitochondria-LD contact” as the following: “Mitochondria-LD contact has been reported in a variety of cell types and tissues although the molecular machinery regulates the contact just begin to be uncovered. Several key players that are known to regulate mitochondria-LD contact including the SANRE protein SNAP23, mitochondrial fusion protein Mfn2 and a mitochondria outer membrane protein mitoguardin 2 (MIGA2) as well as the LD protein perilipin 1 (PLN1) and perilipin 5 (PLN5). SNAP23 may interact with the long chain acyl-CoA synthetase 1 (ACSL1), which is located on the mitochondrial surface, to promote the mitochondria-LD contact. Knockdown of SNAP23 inhibits mitochondria-LD contact and decreases β-oxidation. In brown adipose tissue, PLN1 directly interacts with Mfn2 and ablation of Mfn2 decreases mitochondria-LD contact. PLN5 is also detected on mitochondria-LD contact sites and promotes the contact formation although its interacting partner on mitochondria has not been identified yet. MIGA2 also directly tether mitochondria to LD via its amphipathic LD-targeting motif and promotes de novo lipogenesis in mitochondria and facilitate lipid storage in LDs. The currently known machinery that regulates mitochondria-LD contact is illustrated in Figure 3”.

2. When I saw the electron micrographs of mito-LD contacts (especially Fig. 3B, D or 5A), I had a strong feeling that a part of mitochondria protrudes into LD and thus the contacts look electron-dense. I would like to hear author’s opinion on this view.

Response to Comment #2: Thanks for the comment. This is a very good point although we do not have clear evidence at this moment. Nonetheless, we have made some changes in the text to discuss a little bit more on these possibilities as below:

“It is currently unknown how these previously undescribed electron-dense membrane structures are formed in mitochondria-LD contact sites. It is possible that the membrane structures could be due to the protrusion of mitochondrial membrane into the LDs or de nova synthesis at the contact sites. Future studies are needed to confirm these mitochondrial-LD contacts and determine their origin. Moreover, the pathophysiological significance is also unclear although it may have several possible implications that we like to hypothesize on below”.

3. Are these atypical mito-LD contacts seen only after treatment or starvation? The authors should include micrographs from control (non-treated or mock) hepatocytes to make clear this point.

Response to Comment #3: Thanks for the comment. We checked the EM images that we collected during the past years from several independent experiments. We also found similar mito-LD contact in primary cultured hepatocytes although it seems the frequency is lower. It is currently unclear whether stresses such as ethanol or starvation would further increase the mito-LD contact. This will need more careful and rigorous quantitative EM work for the future, perhaps an original research paper. Thus, we have made changes in the text to mention this with caution to avoid over-interpretation:

“The atypical mito-LD contact was also observed in regular cultured primary hepatocytes although the frequency seemed to be lower than ethanol-treated or starved hepatocytes. It is likely the atypical “mito-LD” contacts would increase in conditions with high energy demand. More careful and rigorous quantitative EM work is needed in the future to further confirm whether the atypical mito-LD contact would be increased in response to specific conditions”.

4. Honestly saying, I could not see any direct physiological link between progression of NAFLD/ALD and mito-LD contacts, although there is a link between NAFLD/ALD and LD formation/beta oxidation, or between LD formation/beta oxidation and mito-LD contacts.

Response to Comment #4: Thanks for the comment. We agree with this reviewer and we have pointed out the implications of mito-LD in NAFLD/ALD still largely at the infancy stage and hopefully this review/perspective paper would stimulate more future works from others to provide more insights.

5. Calcium homeostasis and the role of mito-ER contact have been implicated in the progression of NAFLD. Although it is somewhat depicted in the Fig. 2, I could not find any text mentioning the physiological link between NAFLD and calcium homeostasis. The authors should describe it somewhere in the text.

Response to Comment #5: Thanks for the comment. As suggested, we have added discussion as below:

“Calcium (Ca2+) homeostasis is critical for hepatocyte functions. High concentrations of Ca2+ are stored in the ER (ranges from 100-800 µM) but ER of Ca2+ can be released in response to activation of specific receptors on the ER surface including inositol 1,4,5-trisphosphate receptors (IP3R) and ryanodine receptors (RyR). The released Ca2+ is taken up by mitochondria through voltage-dependent anion chanells (VDACs) in the outer mitochondrial membrane at MAMs. Glucose-regulated protein 75 (Grp75) is a heat shock protein and acts as bridging protein for the physically interactions of IP3R with VDAC. Mitochondrial calcium uniporter (MCU) regulates Ca2+ uptake across the inner mitochondrial membrane into the matrix, which is driven by the negative charge of mitochondrial membrane potential. While the physiological levels of mitochondria Ca2+ is important in regulating mitochondria metabolism, persistent increased mitochondrial Ca2+ levels can lead to opening of the mitochondrial permeability transition pore and apoptosis. Dysregulation of hepatic Ca2+ homeostasis has been linked to NAFLD/NASH as increased expression of IP3R is associated with the degree with steatosis in NASH patients”.

Reviewer 3 Report

In this review article, the authors focused on the possible roles of mitochondria-ER and mitochondria-LD contact sites in hepatocytes and hepatic lipid metabolism. The authors also presented some ultrastructure electron microscopy (EM) images of hepatocytes from their own studies showing a unique mitochondria-LD contact structure under physiological and pathological conditions and discuss its potential implications in lipid metabolism and NAFLD/ALD.

Comments:

This is an interesting review article. The reviewer has some minor concerns as follows:

  1. Insulin resistance has been shown to be a key pathogenic feature of the metabolic syndrome and is recognized as the most common risk factor for NAFLD development and progression. The role of the mitochondrial-associated membrane (MAM) in the control of glucose homeostasis in both health and metabolic diseases has been examined and discussed. Therefore, the discussion for the possible roles of mitochondria-ER and mitochondria-LD contact in hepatic glucose metabolism can be considered.
  2. A recent study by Anastasia et al. has important findings about that mitochondria-rough-ER contacts in the liver regulate systemic lipid homeostasis (Cell Reports, volume 34, issue 11, 2021). It can be considered as a reference.
  3. The authors presented some ultrastructure electron microscopy (EM) images of mouse hepatocytes in Figs. 1, 3, 4, and 5A. It needs to be clearly described whether these data are unpublished or published.
  4. In Figure 1A, the location for ER needs to be indicated.
  5. There are many abbreviations in this manuscript. The whole names for these abbreviations can be clearly described in the figure legends.

Author Response

Reviewer #3:

In this review article, the authors focused on the possible roles of mitochondria-ER and mitochondria-LD contact sites in hepatocytes and hepatic lipid metabolism. The authors also presented some ultrastructure electron microscopy (EM) images of hepatocytes from their own studies showing a unique mitochondria-LD contact structure under physiological and pathological conditions and discuss its potential implications in lipid metabolism and NAFLD/ALD.

Comments:

This is an interesting review article. The reviewer has some minor concerns as follows:

  1. Insulin resistance has been shown to be a key pathogenic feature of the metabolic syndrome and is recognized as the most common risk factor for NAFLD development and progression. The role of the mitochondrial-associated membrane (MAM) in the control of glucose homeostasis in both health and metabolic diseases has been examined and discussed. Therefore, the discussion for the possible roles of mitochondria-ER and mitochondria-LD contact in hepatic glucose metabolism can be considered.

Response to Comment #1: Thanks for the comment. The potential role phospholipid synthesis and MAM in hepatic glucose metabolism is now briefly discussed. There is little information available on mito-LD in hepatic glucose metabolism. Please also see response to Reviewer 1 Response to comment #2.

“The Kennedy pathway enzymes are expressed ubiquitously in eukaryotes to regulate the synthesis of PLs, which are essential to form cellular membranes to protect the cells and make barriers for organelles within the cells. In hepatocytes, dysregulation of PLs synthesis, in particular the abundance of PC and PE as well as the molar ratio of PC/PE can affect the size and dynamics of LDs, assembly and secretion of VLDL as well as mitochondrial bioenergetics. For instance, liver-specific Pcyt1α (encoding the protein CT) knockout mice and whole body Pemt knockout mice fed with a high fat diet have decreased PC contents in hepatocytes and impaired hepatic VLDL secretion. In addition, hepatocytes lack of PEMT have decreased PC/PE molar ratio with increased mitochondrial PE content, which leads to reduced hepatic glucose production from pyruvate, increased accumulation of elongated mitochondria and mitochondrial respiration as well as ATP production. As a result, Pemt knockout mice are resistant to high fat diet-induced obesity and insulin resistance although they still develop NASH.

To date, there is still very limited understanding on the local concentration and actual lipid composition of the MAMs. It has been shown that depletion of cholesterol in MAMs with methyl-β-cyclodextrin (MβC) increases the association of MAMs with mitochondria.  It is likely that there are different proteins and lipid compositions in MAMs in different cell types or in response to different physiological or pathological stresses. It is also perceivable that the changes of the specific lipid contents and combinations within the MAMs may lead to recruit to different sets of proteins and thus the functions of MAMs. Future works are needed to specifically manipulate the lipid contents and compositions in the MAMs to further dissect the role of PLs in regulating MAMs and its subsequent impacts on ER and mitochondrial functions”.

2. A recent study by Anastasia et al. has important findings about that mitochondria-rough-ER contacts in the liver regulate systemic lipid homeostasis (Cell Reports, volume 34, issue 11, 2021). It can be considered as a reference.

Response to Comment #2: Thanks for the comment. We now cited the reference and added discussion as below: “A recent study reported the contact of rough ER with mitochondria in hepatocytes, which is termed as wrapper-mitochondria contact. WrappER-mitochondrial contact site may serve as a site for VLDL biogenesis in the mouse liver, and ablation of the WrappER-mitochondrial contact decreases VLDL secretion resulting in the systemic changes of lipids in mice”.

3. The authors presented some ultrastructure electron microscopy (EM) images of mouse hepatocytes in Figs. 1, 3, 4, and 5A. It needs to be clearly described whether these data are unpublished or published.

Response to Comment #3: Thanks for the comment. All these imaging data here are from the authors lab and not published. We now have clearly indicated in the text/figure legend.

4. In Figure 1A, the location for ER needs to be indicated.

Response to Comment #4: Thanks for the comment. Done as suggested.

5. There are many abbreviations in this manuscript. The whole names for these abbreviations can be clearly described in the figure legends.

Response to Comment #5: Thanks for the comment. Done as suggested.

Round 2

Reviewer 2 Report

The authors have made a great effort to improve the manuscript and responded adequately to my comments.